# Acute Kidney Injury with SARS-CoV-2 Infection in Pediatric Patients Receiving High-Dose Methotrexate Chemotherapy: A Report of Three Cases

**DOI:** 10.3390/children10020331

**Published:** 2023-02-09

**Authors:** Olga Nigro, Cristina Meazza, Elisabetta Schiavello, Veronica Biassoni, Nadia Puma, Luca Bergamaschi, Giovanna Gattuso, Giovanna Sironi, Virginia Livellara, Gabriele Papagni, Maura Massimino

**Affiliations:** 1Pediatric Oncology Unit, Fondazione IRCCS Istituto Nazionale dei Tumori, 20133 Milan, Italy; 2Department of Anesthesia and Critical Care, Fondazione IRCCS Istituto Tumori di Milano, 20133 Milan, Italy

**Keywords:** SARS-CoV-2, COVID-19, methotrexate, pediatric, renal failure, AKI

## Abstract

Background. Methotrexate is renally excreted. HDMTX (high dose-methotrexate)-induced acute kidney injury (AKI) is a non-oliguric decrease in glomerular filtration rate (GFR) heralded by an acute rise in serum creatinine. Moreover, AKI is also a frequent complication of COVID-19. Among our patients treated with HDMTX, some of these developed AKI during SARS-CoV-2 infection. Therefore, we wondered whether our patients’ kidney failure might have been triggered by their underlying SARS-CoV-2 positivity. Methods. Data were collected from the database at the Pediatric Oncology Unit of the Istituto Nazionale dei Tumori in Milan (Italy) regarding patients who matched the following selective criteria: (a) treatment with HDMTX during the pandemic period; (b) SARS-CoV-2 infection during the treatment; (c) development of AKI during HDMTX treatment and SARS-CoV-2 infection. Results. From March 2020 to March 2022, a total of 23 patients were treated with HDMTX; 3 patients were treated with HDMTX during SARS-CoV-2 infection and all 3 developed AKI. Conclusions. Clinical manifestations associated with this virus are many, so we are not yet able to lower our guard and rule out this infection as a cause of clinical manifestations with any certainty.

## 1. Introduction

Acute kidney injury (AKI) is associated with several causes, such as drugs, sepsis, renal surgery, nephrotoxic medications such as contrast media, dehydration, ischemia–reperfusion renal injury or ischemia, and urinary tract obstruction [1,2].

The diagnosis of AKI is based on clinical examination and laboratory analysis. Serum creatinine levels and the estimated glomerular filtration rate (eGFR) are utilized to detect AKI [3,4]. The problem associated with the use of these laboratory tests is that early kidney damage is not accompanied by a significant increase in creatinine values [3].

The main management of early AKI involves preventive strategies, such as optimizing volume hemodynamics and volume status, as well as avoiding nephrotoxins. Instead, the kidney replacement therapy (dialysis) remains the only therapeutic option for severe AKI [5]. Recently, other studies on the treatment of AKI have been published. It was demonstrated the reno-protective action of phosphodiesterase 5 inhibitors (PDE5Is), independently of the AKI type and the agent applied [2].

As previously anticipated, multiple drugs are associated with nephrotoxicity and with different mechanisms of action. Among these are also included some chemotherapeutic agents.

Methotrexate (MTX) is renally excreted, and is one of the drugs associated with AKI. The incidence of severe nephrotoxicity from high dose MTX (HDMTX) is 1.8%; it is fatal in 4% of patients who experience acute, severe nephrotoxicity [6]. The acute HDMTX-induced AKI can be reduced by waiting for simple measures, because it mainly depends on host factors, supportive measures used, and the dose and schedule of HDMTX. For example, hyperhydration and urine alkalinization are mandatory during HDMTX treatment [7,8]. Drug-drug interactions can also contribute to delayed methotrexate excretion and subsequent nephrotoxicity [7]. Agents that compete for renal tubular secretion by interfering with MTX clearance are at greatest risk for causing acute and severe AKI [7,8]. HDMTX-induced AKI is a non-oliguric decrease in GFR heralded by an acute rise in serum creatinine. It has been attributed to crystalline AKI from methotrexate precipitation in renal tubules [9,10]. Infusion of HDMTX over 4 h may increase the risk for nephrotoxicity because of the high peak serum and urinary methotrexate concentrations [11]. The nephrotoxicity results in prolonged exposure to high MTX concentrations. As clinical consequences, patients experience increased toxicity including myelosuppression, mucositis, and hepatic dysfunction, and dermatitis [12,13,14].

In addition, AKI is also a frequent complication of COVID-19 [15].

Among our patients treated with HDMTX, some were infected by SARS-CoV-2 during the infusion. Therefore, we wondered whether our patients’ kidney failure might have been triggered by their underlying SARS-CoV-2 positivity.

## 2. Materials and Methods

Data was collected from the database at the Pediatric Oncology Unit of the Istituto Nazionale dei Tumori in Milan (Italy) regarding patients who matched the following progressively selective criteria: (a) chemotherapeutic treatment with HDMTX during the pandemic period; (b) SARS-CoV-2 infection during the treatment; (c) development of AKI during HDMTX treatment and SARS-CoV-2 infection.

The study was approved by the ethics committee of the Fondazione IRCCS Istituto Nazionale dei Tumori (N. INT 104/22).

The patient medical records were analyzed, evaluating demographic characteristics, clinical and laboratory data. The demographic characteristics considered were gender, age, weight and height. The clinical data collected were comorbidities, any allergic reactions to MTX infusion, vital parameters, MTX toxicities other than AKI. Finally, the laboratory data needed for our analysis were: COVID-19 test results, serum MTX levels (24, 48, 72 h after HDMTX infusion and, if levels still not normalized, also subsequent time intervals), serum creatinine values and blood urea nitrogen (BUN; time 0 (before chemotherapy), 24, 48, 72, 90, 120 h after HDMTX infusion and, if serum creatinine levels still not in range, even subsequent time intervals), aspartate aminotransferase/alanine aminotransferase (AST/ALT; time 0 (before chemotherapy), 24, 48, 72, 90, 120 h after HDMTX infusion and, if serum creatinine levels still not in range, even subsequent time intervals.

The patients considered in our study were all not vaccinated as, at the time of the infection, the COVID-19 vaccine had not been approved in Italy. Furthermore, remdesivir had not yet been approved, so none of the patients was treated with the antiviral.

## 3. Results

The analysis concerned a total of 23 patients (age 2–20 years, median 10) treated with HDMTX at our Unit from March 2020 to March 2022. Of these, 16 were diagnosed with osteosarcoma, 5 with Burkitt lymphoma and 2 with medulloblastoma. In total, 213 patients were infected with SARS-CoV-2 during the pandemic. Three of these were treated with HDMTX during the SARS-CoV-2 infection (1 patient diagnosed with osteosarcoma, 1 with Burkitt lymphoma and 1 with medulloblastoma). All three patients treated with HDMTX during SARS-CoV-2 infection developed AKI.

### 3.1. Patients’ Medical Reports

#### 3.1.1. Patient 1

A 12-year-old boy was diagnosed with localized high-grade osteosarcoma. The first cycle of neoadjuvant chemotherapy included HDMTX 12 g/m^2^, but infusion was stopped immediately due to ubiquitous pomphi, hyperemia, and intense itching. Despite high doses of i.v. hydrocortisone and antihistamines, a new attempt at infusion had to be abandoned after 44% of the total dose, infused in 3.5 h. Several episodes of vomiting and diarrhea occurred in the following hours (co-culture for adenovirus and rotavirus, and Clostridium difficile: negative). After 24 h, serum MTX was 7.33 micromol/L, creatinine was 1.09 mg/dL (previously: 0.5 mg/dL, normal value < 1.2 mg/dL), and BUN was normal. The boy also developed severe hypertension, which gradually improved with amlodipine, continued for two months. After 48 and 72 h, serum MTX was respectively 1.55 and 0.52 micromol/L, and creatinine increased to 1.2 mg/dL. Levels normalized over the next two days (Table 1).

Due to positive tests for SARS-CoV-2 in family members, patient had a nasopharyngeal swab before starting the chemotherapy, testing negative on reverse transcriptase polymerase chain reaction (RT-PCR) assay. The test was repeated the day after the MTX infusion and resulted positive, though the patient never presented COVID-19 symptoms. After 90 days, RT-PCR assay was still positive. Adriamycin and cisplatin were continued during this time without any toxicity, but no HDMTX. Before major surgery, hyperimmune plasma therapy was administered 93 days after the first positive COVID-19 test result, obtaining the negativity on the 103rd day. The boy underwent surgery, and then began adjuvant chemotherapy. Cisplatin was well tolerated. The dose of HDMTX was reduced to 8 g/m^2^, and a premedication (hydrocortisone and antihistamines) was prescribed. Again, within minutes of starting the infusion, he developed a diffuse erythematous rash. I.v. hydrocortisone was administered and MTX infusion was attempted again (33% of the total dose was administered). Creatinine remained within normal range. Serum MTX levels after 24 and 48 h were as expected. The treatment schedule was changed, introducing ifosfamide and omitting all HDMTX cycles.

#### 3.1.2. Patient 2

An 8-year-old boy was admitted to the Neurosurgery Department with a metastatic posterior fossa neoplasm later diagnosed as classical medulloblastoma, non-SHH (Sonic Hedgehog), non-WNT (Wingless-Integrated) molecular subgroup. During his hospital stay, he tested positive for SARS-CoV-2, and subsequent weekly RT-PCR tests were always positive for a month. The child never presented symptoms of COVID-19. A personalized chemotherapy was started with carboplatin 600 mg/m^2^, with no acute side effects. A further COVID-19 test performed before starting a second course of chemotherapy was still positive. He was administered vincristine 1.4 mg/m^2^ + HDMTX 8 g/m^2^. Within a few hours of starting the infusion, he developed significant nausea and numerous episodes of vomiting. Fever prompted blood cultures (negative) and antibiotic therapy was started. Blood tests for adenovirus, Cytomegalovirus, Epstein-Barr virus, Hepatitis A virus, Hepatitis C virus, Hepatitis B virus, and toxoplasma were negative. Chest X-ray revealed no pneumonia. Post-MTX blood tests showed an increase in creatinine (0.6 mg/dL, double the baseline value) and transaminases (AST/ALT 2633/3418 IU/mL), with a reduced MTX clearance (30 mmol/l after 24 h). I.v. liquid infusion was increased and calcium levofolinate administered. Over the following days, creatinine gradually returned to normal, transaminase levels dropped on the third day post-MTX, and serum MTX levels also decreased (0.12 mmol/l after 90 h) (Table 1, Patient 2). Sixty days later, the boy was still testing positive for SARS-CoV-2. Chemotherapy was continued without MTX (Table 2).

#### 3.1.3. Patient 3

In October 2021, an 8-year-old boy was diagnosed with Burkitt lymphoma, for which he started chemotherapy, which was well tolerated, without significant side effects. At the end of December 2021, due to fever and cough, he performed a RT-PCR assay for SARS-CoV-2, testing positive. Chest X-ray and Computed Tomography (CT)-scan were negative. During the positivity for COVID-19 we administered chemotherapy with i.v. etoposide, without side effects. After 20 days from the first molecular swab, the patients still tested positive for SARS-CoV-2. He started chemotherapy with HD-MTX 250 mg/kg and vincristine 1.4 mg/m^2^ and, in a few hours, presented several emetic episodes. Serum MTX levels after 24 h were very high (78.94 micromol/L), and the creatinine was 1.02 mg/dL (previously: 0.30 mg/dL). On the third day, we witnessed an increase in transaminase values (AST/ALT 553/1674 IU/L). I.v. liquid infusions were consequently increased and calcium levofolinate was administered, with a gradual decrease of serum MTX levels, which was still dosable on the fifth day (0.48 micromol/L), and an initial reduction in creatinine (0.74 mg/dL) and AST/ALT levels (807/1552 IU/L). Levels normalized over the next five days (Table 1, Patient 3). Blood pressure values at the upper limits of normal were also reported. After 28 day the boy still had a positive RT-PCR swab for SARS-CoV-2 (Table 3).

## 4. Discussion

AKI induced by HDMTX does not appear to depend on the total dose administered, and the above three clinical cases reinforce this hypothesis. All our patients experienced vomiting and diarrhea. Only the first patient’s AKI was accompanied by hypertensive peaks; the first and third patients experienced also hypertransaminasemia.

As AKI is also a frequent complication of COVID-19, we wondered whether our patients’ kidney failure might have been triggered by their underlying SARS-CoV-2 positivity. Judging from published reports on COVID-19-related AKI [15], it is presumably of multifactorial origin, and involves direct viral infestation of the kidneys, as well as acute tubular necrosis [16]. Reports mention SARS-CoV-2 viral particles in renal tubular cells and podocytes. There seems to be a co-expression of angiotensin-converting enzyme 2 (ACE2) and Transmembrane Serine Proteases (TMPRSSs) genes in podocytes and proximal tubules that makes them a potential host for SARS-CoV-2 [17,18]. Zaim et al. showed that 0.5–19% of COVID-19 patients demonstrate AKI [19].

At the time of infection, our patients were not vaccinated (as the vaccination had not yet been approved in Italy in the pediatric field). Canney et al. reported that in a population-level cohort of patients with glomerular disease, a second or third dose of COVID-19 vaccine was associated with higher relative risk but low absolute increased risk of relapse [20]. At the time of SARS-CoV-2 infection in our patients, remdesivir had not yet been approved in Italy, so none of our patients were treated with the antiviral. From recent studies, remdesivir can be tried in moderate-to-severe COVID-19 cases with renal dysfunction as a complete recovery of renal function was noted in survivors [21,22]. Our patients’ MTX toxicity was unusual. In the first case, non-pathological MTX levels were wholly or partially responsible for AKI and hypertension. The second and third patients had toxic levels of MTX, but they rapidly decreased, leading to very short-lived renal damage; the second patient also had grade 4 (G4) liver toxicity. The first patient also showed an allergic reaction to MTX, which raises another question. Type I hypersensitivity reactions are mediated by Immunoglobuline E (IgE), occurring rapidly (minutes or hours) and sometimes resulting in anaphylaxis or angioedema.

Could such an allergic hypersensitivity reaction be partly linked to the cytokine storm triggered by COVID-19? Several groups reported an association between human herpes virus 6 (HHV-6) reactivation and certain conditions, such as drug reactions with eosinophilia and systemic symptoms (DRESS). Viral infections have also been implicated in mimicking, triggering, or worsening cases of urticaria. COVID-19 patients reported drug hypersensitivity (11%) and urticaria (1.4%) as well [10]. Various drugs used in different stages of COVID-19 seem to cause rare but potentially severe hypersensitivity reactions [23,24,25].

Given our suspicion of a transient hypersensitivity reaction, MTX was reintroduced, after the negativity for COVID-19. Unfortunately, we had again a skin hypersensitivity reaction.

Another important point common to MTX-treated patients is the persistence of SARS-CoV-2 infection in the absence of clinical symptoms; an important question is how to deal with pediatric cancer patients requiring chemotherapy who are positive for SARS-CoV-2 without COVID-19 symptoms. In fact, in our institution, anticancer agents such as carboplatin, cisplatin, and vincristine were commonly used at the time of SARS-CoV-2 infection, but no unexpected toxicity was observed. On the other hand, data on the risks and effects of cytotoxic therapy during SARS-CoV-2 positivity are limited and little is known about the adult setting [26,27,28,29,30,31,32,33,34,35].

In the literature, there are currently no data available to help in choosing whether to administer chemotherapeutic agents during SARS-CoV-2 infection (symptomatic or asymptomatic). In our experience, HDMTX can cause AKI and other types of unexpected toxicities in both symptomatic and asymptomatic COVID-19 patients. Therefore, based on our experience, during SARS-CoV-2 infection in both asymptomatic and symptomatic patients, we subsequently avoided the administration of HDMTX. Based on this experience alone, we absolutely do not feel able to demonstrate that the use of HDMTX or other anticancer drugs with risk of renal damage should be avoided during SARS-CoV-2 infection. This is a very limited case series, so it is difficult to draw conclusions; studies with a much larger number of patients are needed to confirm what has been observed at our institution.

Moreover, in our series, at our institute other anticancer agents were commonly used in the course of SARS-CoV-2 infection, without inducing unexpected toxicities (carboplatin, cisplatin, vincristine and others).

The role of immune suppression in COVID-19 is unclear. Small doses of dexamethasone lower the associated mortality rate by stopping autoimmune destruction of the lungs because of the so-called cytokine storm [26,27,28]. On the other hand, several reports indicate that immune suppression in early stages of COVID-19 is dangerous. Another drug considered useful in preventing aggressive autoimmune reactions is cyclophosphamide, and it has been used in efforts to prevent cytokine storms [27,28]. There are limited, but consistent data suggesting that other immunosuppressants, such as rituximab, can worsen the course of COVID-19. Rituximab causes peripheral B lymphocyte depletion, which leads to severe humoral and cellular immunodeficiencies. Some cases have been reported of persistent SARS-CoV-2 viremia in patients given rituximab treatment [31,32]. Rituximab can also prolong the SARS-CoV-2 incubation period [33,34,35]. The third patient of our study was also supposed to be treated with Rituximab but, given the SARS-CoV-2 infection, the drug was not administered. However, he had been treated with Rituximab in the period prior to the infection.

## 5. Conclusions

Almost three years after the start of the COVID-19 pandemic, there are still many questions to be answered. We have only begun to study this virus, and there is a long way to go, but much progress has already been made in our understanding of its effects. Unfortunately, clinical manifestations associated with it are many, so we are not yet able to lower our guard and rule out this infection as a cause of clinical manifestations with any certainty.

Large case-control and prospective studies are needed in order to confirm the hypotheses advanced in this study.

## Figures and Tables

**Table 1 children-10-00331-t001:** Baseline, 24 h, 48 h and 72 h values (nv = normal values; h = hours).

	Gender	Age (Years)	Weight (Kg)	Height (cm)	MTX Dosage	Serum MTX (Micromol/L)	Serum Creatinine Values (mg/dL)	AST/ALT (IU/L)
Day 0	Male	12	40	130	12 g/m^2^ (44% of the total dose admnistered)	-	0.5	20/10
+24 h					-	7.33	1.09	-
+48 h					-	1.55		27/35
+72 h					-	0.52		nv

**Table 2 children-10-00331-t002:** Baseline, 24 h, 48 h, 72 h, and 90 h values (nv = normal values; h = hours).

	Gender	Age (Years)	Weight (Kg)	Height (cm)	MTX Dosage	Serum MTX (Micromol/L)	Serum Creatinine Values (mg/dL)	AST/ALT (IU/L)
Day 0	Male	8	35.3	150	8 g/m^2^	-	0.3	20/18
+24 h					-	30	0.6	2633/3418
+48 h					-	-		-
+72 h					-	-		nv
+90 h					-	0.12		nv

**Table 3 children-10-00331-t003:** Baseline, 24 h, 48 h, 72 h, 90 h, 120 h, and 10 days values (nv = normal values; h = hours).

	Gender	Age (Years)	Weight (Kg)	Height (cm)	MTX Dosage	Serum MTX (Micromol/L)	Serum Creatinine Values (mg/dL)	AST/ALT (IU/L)
Day 0	Male	8	29.2	141	250 mg/Kg	-	0.3	25/22
+24 h					-	78.94	1.02	-
+48 h					-	1.7	0.72	553/1674
+72 h					-	-	0.83	201/1150
+90 h					-	0.64	0.66	936/1581
+120 h					-	0.48	0.74	807/1522
+10 days					-	0.0	0.35	nv

## Data Availability

Reported data are available in the records of patients treated at Fondazione IRCCS Istituto Nazionale dei Tumori di Milano.

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
