# Peer review of "Acute Kidney Injury with SARS-CoV-2 Infection in Pediatric Patients Receiving High-Dose Methotrexate Chemotherapy: A Report of Three Cases"

_children, 2023, doi:10.3390/children10020331_

Round 1

Reviewer 1 Report

Thanks for your manuscript. I think it is an important association that need to be considered in this period. Have you lab values such ast/alt, c3, c4 and urinary electrolytes? 

Author Response

For no patient, unfortunately, we have available c3, c4 and urinary electrolytes.

For the first and second patients ast/alt were normal.

As reported in the text, the second patient showed incremented values of transaminases (AST/ALT 2633/3418 IU/ml), rapidly deacreased (transaminase levels dropped on the third day post-MTX). We added for the third patients AST/ALT values: "On the third day, we witnessed an increase in transaminase values (AST/ALT 553/1674 IU/L)".

Reviewer 2 Report

Title Change to:  Acute kidney injury with SARS-CoV-2 infection in pediatric patients receiving high-dose methotrexate chemotherapy?

Major Comments

Please indicate for each case if they had received COVID-19 vaccination and cite the literature by Canney et al on vaccination worsening the progression of kidney disease

Please indicate for each case if they had received remdesiver and cite the literature on the contribution of remdesiver to AKI

Please disclose early ambulatory treatment for COVID-19 in the prehospital phase for each case.  If there was no prehospital treatment given then please cite this as a shortcoming in the care of patients.  Please comment that prehospital treatment is the only method by which a hospitalizaton can be avoided in an acutely ill patient.  Cite:  https://www.mdpi.com/2673-8112/2/8/84

Author Response

No patients had received the COVID vaccination, as the infection arose before the vaccine was approved in Italy in the pediatric field. I cited Canney et al. in the text (yellow).

Similarly, no patient was treated with remdesiver as it was not yet approved in Italy at the time of infection in these patients. I added a paragraph of the importance of remdesiver on AKI in the text (yellow).

All patients were already hospitalized when SARS-CoV-2 infection was discovered, therefore it was not possible to speak about the early ambulatory treatment for COVID-19 in the prehospital phase.

I changed the title into: Acute kidney injury with SARS-CoV-2 infection in pediatric patients receiving high-dose methotrexate chemotherapy. (yellow)

Reviewer 3 Report

1. The manuscript is actually a compilation of 3 case reports. This has to be clearly stated in the title. 

2. A table should be prepared including the baseline values, the intermediate values and the recovery values, demographics and dosages for all case reports in order for the reader to be able to compare easily.

2. An in-depth literature review should be briefly presented on AKI, too, and its connection with medication along with prevention strategies (eg J Clin Med. 2020 Apr 29;9(5):1284. doi: 10.3390; Pharmacol Ther. 2017 Dec;180:99-112. doi: 10.1016; etc).

3. In addition, the present observations should be discussed with regards to common knowledge with regards to AKI and not only with regards to covid-19 infection. 

4. More statistical data should be provided on AKI prevalence after methotraxate treatment or treatrment with other similar substances even in adults,but also in a pediatric population, and discuss the characterittics of those prone to AKI.

5. Editorial checking throughout the manuscript is recommended and especially in the abstract where there are a lot of abbreviations without being explained.

Author Response

  1. We changed the title into: "Acute kidney injury with SARS-CoV-2 infection in pediatric patients receiving high-dose methotrexate chemotherapy. A report of three cases."
  2. A table with the baseline values, the intermediate values and the recovery values, demographics and dosages for all case reports was added.
  3. We inserted in the introduction a literature review on AKI and its connection with medication along with prevention strategies (green)
  4. We treated AKI not only associated with COVID and MTX, but in general (green).
  5. We added more statistical data should be provided on AKI prevalence after methotraxate treatment (green).
  6. Abbreviations were explained in the abstract and the text.

Round 2

Reviewer 3 Report

The authors have done a great job addressing the vast majority of the comments in an appropriate way. One minor revision is still left to be made  before acceptance for publication:

The authors should discuss the low rebound creatinine values which are consistent both in animal and human reports on AKI after contrast media administration.

Author Response

Thanks to the reviewer for the clarification. That would be a great point to analyze, though none of our patients received contrast media in the 3 weeks prior to HDMTX treatment and COVID infection. Contrast-induced nephrotoxicity was not discussed separately, as the authors chose to focus exclusively on MTX- and COVID-induced AKI. The causes of AKI, other than contrast media, have not been deliberately investigated either.